# OPENSTEREO: A COMPREHENSIVE BENCHMARK FOR STEREO MATCHING AND STRONG BASELINE

## ABSTRACT

Stereo matching aims to estimate the disparity between matching pixels in a stereo image pair, which is important to robotics, autonomous driving, and other computer vision tasks. Despite the development of numerous impressive methods in recent years, determining the most suitable architecture for practical application remains challenging. To address this gap, our paper introduces a comprehensive benchmark focusing on practical applicability rather than solely on individual models for optimized performance. Specifically, we develop a flexible and efficient stereo matching codebase, called **OpenStereo**. OpenStereo includes training and inference codes of more than 10 network models, making it, to our knowledge, the most complete stereo matching toolbox available. Based on OpenStereo, we conducted experiments and have achieved or surpassed the performance metrics reported in the original paper. Additionally, we conduct an exhaustive analysis and deconstruction of recent developments in stereo matching through comprehensive ablative experiments. These investigations inspired the creation of **StereoBase**, a strong baseline model. Our StereoBase ranks $1^{st}$ on SceneFlow, KITTI 2015, 2012 (Reflective) among published methods and performs best across all metrics. In addition, StereoBase has strong cross-dataset generalization.

## 1 INTRODUCTION

Stereo matching is a fundamental topic in the field of computer vision, aiming to compute the disparity between a pair of rectified stereo images. It plays a crucial role in numerous applications such as robotics Zhang et al. (2015), autonomous driving Mur-Artal & Tardós (2017); Shamsafar et al. (2022), and augmented reality Yang et al. (2020), as it enables depth perception and 3D reconstruction of the observed scene.

Traditional stereo-matching algorithms typically match corresponding image regions between the left and right views based on their similarity measures. Several techniques have been proposed in the literature for stereo matching, including methods based on gray-level information Birchfield & Tomasi (1999); Li & Wu (2013); Yang et al. (2010), region-based approaches Zhang & Kosecka (2005); Pinggera et al. (2015), and energy optimization methods Scharstein & Szeliski (2002); Hirschmuller (2007). With the support of large synthetic datasets Mayer et al. (2016); Scharstein et al. (2014); Schops et al. (2017); Geiger et al. (2012); Menze & Geiger (2015); Yang et al. (2019b), CNN-based stereo matching methods Kendall et al. (2017); Chang & Chen (2018); Guo et al. (2019); Xu et al. (2023) has achieved impressive results. As shown in Figure 1, based on the network pipeline of stereo matching, CNN-based stereo matching methods can be roughly grouped into two categories Wang et al. (2021), including the encoder-decoder network with 2D convolution (ED-Conv2D) Mayer et al. (2016); Poggi et al. (2019); Yang et al. (2019a); Saikia et al. (2019); Wang et al. (2021); Xu & Zhang (2020); Tosi et al. (2021); Li et al. (2021); Lipson et al. (2021); Li et al. (2022); Weinzaepfel et al. (2023); Li et al. (2024) and the cost volume matching with 3D convolution (CVM-Conv3D) Kendall et al. (2017); Chang & Chen (2018); Zhang et al. (2019); Guo et al. (2019); Zhang et al. (2020a); Duggal et al. (2019); Zhang et al. (2020b); Gu et al. (2020); Badki et al. (2020); Cheng et al. (2020); Bangunharcana et al. (2021); Shen et al. (2021); Xu et al. (2022; 2023); Chen et al. (2024); Xu et al. (2024).

However, we find that different studies often employ various data augmentation strategies, learning rates, learning rate optimization methods, and backbone architectures. This inconsistency makes it

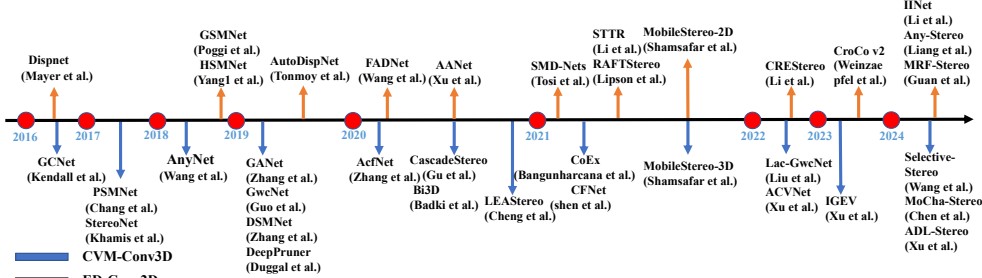

Figure 1: **Timeline of Stereo Matching Models.** The top part shows ED-conv2D-based models, while the bottom part shows CVM-conv3D-based models. Each model is labeled with its name and authors.

difficult to evaluate and compare various methods' performance accurately. This inconsistency in experimental setups and methodologies makes it difficult to derive conclusive insights and hampers the objective assessment of advancements in stereo matching. Without a standardized benchmark, researchers struggle to identify the true impact of individual components and innovations. There is a lack of clear conclusions and exploration regarding data augmentation strategies, backbone selection, and cost construction methods in stereo matching.

Moreover, not all datasets are accompanied by official evaluation tools. For example, the Driving-Stereo Yang et al. (2019b) dataset does not provide specific evaluation scripts, making comparative assessments challenging. The SceneFlow Mayer et al. (2016) dataset, with its finalpass and cleanpass data varieties, complicates fair model comparisons. Generalization experiments for stereo matching algorithms typically train on the SceneFlow dataset and evaluate on KITTI2012 Geiger et al. (2012), KITTI2015 Menze & Geiger (2015), ETH3D Schops et al. (2017), and Middlebury Scharstein et al. (2014). However, due to the absence of a standard protocol for generalization experiments, different papers may yield inconsistent results for the same method. For instance, discussions on the generalization performance of IGEV Xu et al. (2023) across works Xu et al. (2023); Wang et al. (2024b); Guan et al. (2024) exemplify this issue.

Hence, there's a pressing need for a comprehensive benchmark study within the stereo-matching community to enhance practicality and ensure consistent comparisons. To achieve this objective, we introduce a versatile stereo-matching codebase: OpenStereo. To promote scalability and adaptability, OpenStereo offers the following features: (1) **Modular design**, researchers can define a new model without the need to alter the model code itself by simply modifying a YAML configuration file. (2)**Various frameworks**, including Concatenation-based Chang & Chen (2018); Zhang et al. (2019), Correlation-based Xu & Zhang (2020); Wang et al. (2020; 2021); Bangunharcana et al. (2021), Interlaced-based Shamsafar et al. (2022), Group-wise-correlation-based Xu et al. (2023), Combine-based methods Guo et al. (2019), and Difference-based Khamis et al. (2018). (3)**Various datasets**, including SceneFlow Mayer et al. (2016), KITTI2012 Geiger et al. (2012), KITTI2015 Menze & Geiger (2015), Middlebury Scharstein et al. (2014), ETH3D Schops et al. (2017) and DrivingStereo Yang et al. (2019b) dataset. (4) **State-of-the-art methods**, including PSM-Net Chang & Chen (2018), GwcNet Guo et al. (2019), AANet Xu & Zhang (2020), FADNet++ Wang et al. (2021), CFNet Shen et al. (2021), STTR Li et al. (2021), CoEx Bangunharcana et al. (2021), CascadeStereo Gu et al. (2020), MobileStereoNet Shamsafar et al. (2022) and IGEV Xu et al. (2023).

Leveraging OpenStereo, we rigorously reassess various officially stated conclusions by re-implementing the ablation studies, including data augmentation, backbone architectures, cost construction, disparity regression, and refinement processes. Based on the insights gleaned from these ablation experiments, we introduce StereoBase, a model that sets a new benchmark, surpassing recently proposed methods in terms of performance. StereoBase is powerful and serves as an empirically state-of-the-art (SOTA) baseline model for stereo matching, demonstrating exceptional efficacy and resilience across diverse testing scenarios. Our contributions are summarized as follows:

- We introduce **OpenStereo**, a unified and extensible platform, which enables researchers to conduct comprehensive stereo matching studies.

- We conduct a profound revisitation and thorough deconstruction of recent stereo-matching methodologies.

- We introduce **StereoBase**, which sets a new benchmark with EPE (End-Point Error) of 0.34 on SceneFlow Mayer et al. (2016) and ranks 1$^{st}$ on KITTI2015 Menze & Geiger (2015) and 2012(Reflective)Geiger et al. (2012) leaderboards among published methods.

## 2 RELATED WORK

### 2.1 STEREO MATCHING

With the rapid development of CNNs, significant progress has been made in stereo matching. Based on the network pipeline of stereo matching, stereo matching methods can be roughly grouped into two categories Wang et al. (2021), including the encoder-decoder network with 2D convolution (ED-Conv2D) and the cost volume matching with 3D convolution (CVM-Conv3D).

**Stereo Matching with CVM-Conv3D** The CVM-Conv3D methods are proposed to improve the performance of depth estimation Kendall et al. (2017); Chang & Chen (2018); Yang et al. (2018); Wang et al. (2019); Zhang et al. (2019); Guo et al. (2019); Zhang et al. (2020a); Duggal et al. (2019); Zhang et al. (2020b); Gu et al. (2020); Badki et al. (2020); Cheng et al. (2020); Bangunharcana et al. (2021); Shen et al. (2021); Xu et al. (2022; 2023; 2024). These methods learn disparities from a 4D cost volume, mainly constructed by concatenating left feature maps with their corresponding right counterparts across each disparity level Chang & Chen (2018). GCNet Kendall et al. (2017) firstly introduced a novel approach that combines 3D encoder-decoder architecture with a 2D convolutional network to obtain a dense feature representation, which is used to regularize a 4D concatenation volume. Following GCNet Kendall et al. (2017), PSMNet Chang & Chen (2018) proposes an approach for regularizing the concatenation volume by leveraging a stacked hourglass 3D convolutional neural network in tandem with intermediate supervision. To enhance the expressiveness of the cost volume and ultimately improve performance in ambiguous regions, GwcNet Zhang et al. (2019) proposes the group-wise correlation volume and ACVNet Xu et al. (2022) proposes the attention concatenation volume. CoEx Bangunharcana et al. (2021) proposes a novel approach called Guided Cost volume Excitation (GCE), which leverages image guidance to construct a simple channel excitation of the cost volume. IGEV-Stereo Xu et al. (2023) leverages an iterative geometry encoding volume to capture local and non-local geometry information, outperforming existing methods on KITTI benchmarks and achieving cross-dataset generalization and high inference efficiency.

However, these CVM-Conv3D methods still suffer from low time efficiency and high memory requirements, which are far from real-time inference even on server GPUs. Therefore, it is essential to address the accuracy and efficiency problems for real-world applications.

**Stereo Matching with ED-Conv2D** The ED-Conv2D methods Mayer et al. (2016); Poggi et al. (2019); Yang et al. (2019a); Saikia et al. (2019); Wang et al. (2020; 2021); Xu & Zhang (2020); Tosi et al. (2021); Li et al. (2021); Lipson et al. (2021); Shamsafar et al. (2022); Li et al. (2022); Weinzaepfel et al. (2023); Li et al. (2024); Guo et al. (2024), which adopt networks with 2D convolutions to predict disparity, has been driven by the need for improved accuracy, computational efficiency, and real-time performance. One of the early deep learning-based stereo matching methods, MC-CNN (Matching Cost CNN) Zbontar et al. (2016), was proposed to learn a matching cost function for improving performance in the cost aggregation and optimization stages. Subsequently, Mayer *et al* Mayer et al. (2016) present end-to-end networks for the estimation of disparity, called DispNet, which is pure 2D CNN architectures. However, the model still faces challenges in capturing the matching features, resulting in poor estimation results. To overcome this challenge, the correlation layer is introduced in the end-to-end architecture Mayer et al. (2016); Dosovitskiy et al. (2015); Ilg et al. (2017; 2018) to better capture the relationship between the left and right images. By incorporating this layer, the accuracy of the model is significantly improved. Furthermore, FAD-Net++ Wang et al. (2021) proposes an innovative approach to efficient disparity refinement using residual learning He et al. (2016) in a coarse-to-fine manner. AutoDispNet Saikia et al. (2019) applied neural architecture search to automatically design stereo-matching network structures. More recently, Croco-Stereo Weinzaepfel et al. (2023) shows that large-scale pre-training can be successful for stereo matching through well-adapted pretext tasks. This method can achieve state-of-the-art performance without using task-specific designs, like correlation volume or iterative estimation.

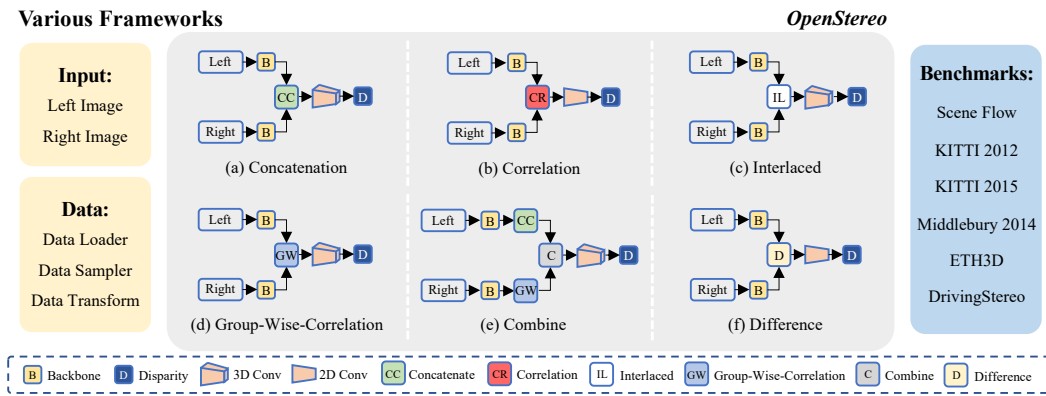

Figure 2: **The design principles of proposed codebase OpenStereo.**

These works represent the significant progress that has been made in the field of stereo matching, highlighting the diverse range of methods and architectures that have been proposed to address the challenges associated with this problem.

## 2.2 CODEBASE

Numerous infrastructure code platforms have been developed in the deep learning research community, with the aim of facilitating research in specific fields. One such platform is OpenGait Fan et al. (2023), a gait recognition library. OpenGait thoroughly examines the latest advancements in gait recognition, providing novel perspectives for subsequent research in this domain. In object detection, MMDetection Chen et al. (2019) and Detectron2 Wu et al. (2019) have emerged as an all-encompassing resource for several favored detection techniques. In pose estimation, OpenPose Cao et al. (2019) has developed the first open-source system that operates in real-time for detecting the 2D pose of multiple individuals, including the detection of key points for the body, feet, hands, and face. In stereo matching, it is noteworthy that not all datasets are accompanied by official evaluation tools. For instance, the DrivingStereo Yang et al. (2019b) dataset does not have official evaluation codes, and there is a lack of unified tools for assessing the model generalization across different domains. This absence of standardized evaluation resources contributes to the observed discrepancies in cross-domain evaluations of the same model as reported in different studies. Therefore, it is time to build a benchmark for stereo matching.

## 3 OPENSTEREO

In recent years, there has been a proliferation of new frameworks and evaluation datasets for stereo-matching. However, the lack of a unified and fair evaluation platform in this field is a significant issue that cannot be ignored. To address this challenge and promote academic research and practical application we have developed **OpenStereo**, a pyTorch-based Paszke et al. (2019) toolbox that provides a reliable and standardized evaluation framework for stereo matching.

### 3.1 DESIGN PRINCIPLES OF OPENSTEREO

As shown in Figure 2, our developed OpenStereo covers the following highlight features.

**Modular Design.** OpenStereo adopts a modular design, greatly facilitating researchers in exploring new networks. By simply modifying a YAML configuration file, researchers can define a new model without the need to alter the model code itself. This design significantly lowers the barriers for researchers to extend or integrate additional algorithms and modules within the framework. This approach empowers researchers to freely combine and customize their algorithms with minimal code composition, enhancing the framework's usability and adaptability.

**Compatibility with various frameworks.** Currently, more and more stereo matching methods have emerged, such as Concatenation-based Kendall et al. (2017); Chang & Chen (2018); Zhang et al.

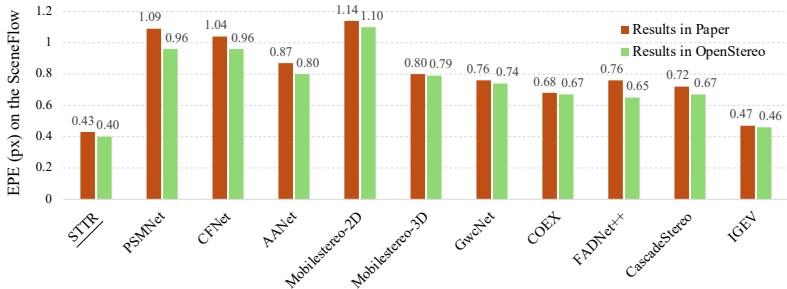

Figure 3: **Quantitative evaluation on the SceneFlow Mayer et al. (2016) and KITTI2015 Menze & Geiger (2015) leadboard.** For each model, the specific category on the SceneFlow used is consistent with the original paper. Underline refers to evaluation in the non-occluded regions only STTR Li et al. (2021).

(2019); Cheng et al. (2020), Correlation-based Xu & Zhang (2020); Wang et al. (2020; 2021); Bangunharcana et al. (2021), Group-wise-correlation-based Xu et al. (2023), Difference-based Khamis et al. (2018), Interlaced-based Shamsafar et al. (2022), and Combine-based methods Guo et al. (2019); Shen et al. (2021). As mentioned above, many open-source methods have a narrow focus on their specific models, making it challenging to extend to multiple frameworks. However, OpenStereo provides a solution to this problem by supporting all of the aforementioned frameworks. With OpenStereo, researchers and practitioners can easily compare and evaluate different stereo-matching models under a standardized evaluation protocol.

**Support for various evaluation datasets.** OpenStereo is a comprehensive tool that not only supports synthetic stereo datasets such as SceneFlow Mayer et al. (2016), but also five real-world datasets: KITTI2012 Geiger et al. (2012), KITTI2015 Menze & Geiger (2015), ETH3D Schops et al. (2017), Middlebury Scharstein et al. (2014), and DrivingStereo Yang et al. (2019b) (More details in the Supplementary Material). We introduce a suite of bespoke functions, meticulously crafted for each dataset, encompassing everything from initial data preprocessing to the final stages of evaluation. The evaluation module includes the submission of the results to KITTI2012 Geiger et al. (2012) and KITTI2015 Menze & Geiger (2015) leadboards.

**Support for state-of-the-arts.** In our work, we have successfully replicated various state-of-the-art stereo matching methods, including PSMNet Chang & Chen (2018), GwcNet Guo et al. (2019), AANet Xu & Zhang (2020), FADNet++ Wang et al. (2021), CFNet Shen et al. (2021), STTR Li et al. (2021), CoEx Bangunharcana et al. (2021), CascadeStereo Gu et al. (2020), MobileStereoNet Shamsafar et al. (2022) and IGEV Xu et al. (2023). As shown in Figure 3, the performance metrics we achieved, in most cases, surpass those reported in their original publications.

## 4 REVISIT DEEP STEREO MATCHING

### 4.1 EVALUATION OF PRIOR WORK

For benchmarking, it is critical to ensure that the results are reliable and trustworthy. To achieve this, we conducted our experiments on SceneFlow Mayer et al. (2016) and KITTI2015 Menze & Geiger (2015) datasets. As shown in Figure 3, the reproduced performances of OpenStereo are better than the results reported by the original papers. (More details in the Supplementary Material). Regarding the KITTI2015 dataset, submission constraints led us to limit our leaderboard contributions to reproductions of the widely recognized PSMNet Chang & Chen (2018) and the latest state-of-the-art IGEV Xu et al. (2023). OpenStereo is designed to offer the research community in stereo matching a standardized, comprehensive platform for method assessment. This facility enables meaningful and comparative analyses across various models.

### 4.2 NECESSITY OF COMPREHENSIVE ABLATION STUDY

In the evolving landscape of deep stereomatching, comprehensive ablation studies play a pivotal role in deciphering the effectiveness of different components and strategies. A thorough ablation study goes beyond mere performance metrics; it uncovers the underlying mechanics of different

Table 1: **Ablation study on SceneFlow Mayer et al. (2016) – Data Augmentation and LR_scheduler Selection.** KITTI2015 Menze & Geiger (2015) training set, consisting of 200 images, is only employed to evaluate the generalizability of models. RC stands for RandomCrop Krizhevsky et al. (2012). HFlip Krizhevsky et al. (2012) denotes both images of a stereo and disparity are horizontally flipped. HSFlip Krizhevsky et al. (2012) horizontally flips both images in the stereo pair and then swaps them. VFlip Krizhevsky et al. (2012) involves vertically flipping both images in the stereo pair along with the disparity, inverting their top-bottom orientation. CES represents ColorAug Krizhevsky et al. (2012), Erase Zhong et al. (2020), and Scale Simonyan (2014). Settings used in our final model are underlined.

| Data Augmentation | LR_scheduler | SceneFlow EPE(px) | KITTI15 EPE(px) | KITTI15 D1_all |
|---|---|---|---|---|
| RC(320×736) | MultiStepLR | 0.6839 | 2.91 | 15.73 |
| RC(320×736) | OneCycleLR | 0.6155 | 2.34 | 11.86 |
| RC(256×512) | OneCycleLR | 0.6470 | 3.02 | 14.95 |
| RC(320×736) | OneCycleLR | 0.6155 | 2.34 | 11.86 |
| RC(320×736)+Scale | OneCycleLR | 0.6867 | 2.88 | 12.91 |
| RC(320×736)+HFlip | OneCycleLR | 0.6612 | 2.22 | 12.27 |
| RC(320×736)+ColorAug | OneCycleLR | 0.6529 | 1.68 | 7.89 |
| RC(320×736)+VFlip | OneCycleLR | 0.6367 | 2.09 | 10.11 |
| RC(320×736)+Erase | OneCycleLR | 0.6167 | 2.68 | 12.64 |
| RC(320×736)+HSFlip | OneCycleLR | **0.6076** | 2.74 | 13.99 |
| RC(320×736)+CE | OneCycleLR | 0.6486 | 1.65 | 8.15 |
| RC(320×736)+CES+HSFlip | OneCycleLR | 0.7165 | 1.71 | 8.40 |
| RC(320×736)+CES | OneCycleLR | 0.7240 | **1.56** | **7.64** |

Table 2: **Ablation study on SceneFlow Mayer et al. (2016) – Backbones Selection**. Flops and Params represent the computational complexity and parameters within the whole model, respectively. FLOPs are calculated at a resolution of $544 \times 960$.

| Backbone | Type | Pretrain | Flops | Params | EPE |
|---|---|---|---|---|---|
| MobilenetV2 100 Sandler et al. (2018) | CNN | | 70.58G | 2.78M | 0.7737 |
| MobilenetV2 100 Sandler et al. (2018) | CNN | ✓ | 70.58G | 2.78M | **0.6155** |
| MobilenetV2 100 Sandler et al. (2018) | CNN | ✓ | **70.58G** | **2.78M** | 0.6155 |
| MobilenetV2 120d Sandler et al. (2018) | CNN | ✓ | 85.93G | 5.21M | 0.5573 |
| EfficientNetV2 Tan & Le (2021) | CNN | ✓ | 157.52G | 24.92M | 0.5207 |
| RepViT Wang et al. (2024a) | Trans. | ✓ | 101.35G | 5.64M | 0.5858 |
| MPViT Lee et al. (2022) | CNN&Trans. | ✓ | 283.35G | 13.33M | **0.5113** |

algorithms, revealing their strengths and weaknesses in various scenarios. For instance, different data augmentation techniques may yield contrasting effects on the model's ability to match stereo images accurately. Similarly, the impact of various backbones, cost volume configurations, and disparity regression methods on the overall performance can be profound. Understanding the specific contributions of each component is crucial for building more efficient and effective stereo-matching systems.

### 4.3 DENOISING STEREO MATCHING

With the support of OpenStereo, a comprehensive reevaluation of various stereo-matching methods is conducted, including data augmentation, feature extraction, cost construction, disparity prediction, and refinement. Our ablation studies have revealed some new insights.

### 4.3.1 LR_SCHEDULER AND DATA AUGMENTATION

As shown in Table 1, MultiStepLR yields an EPE of 0.6839, while OneCycleLR achieves a lower EPE of 0.6155. This substantial difference underscores the crucial role of selecting an appropriate learning rate scheduler for stereo matching. The superior performance of OneCycleLR indicates its potential to improve model accuracy and robustness, making it a preferable choice over MultiStepLR

Table 3: **Ablation study on SceneFlow Mayer et al. (2016) –Cost Construction**. Gwc represents Group-wise correlation volume Guo et al. (2019). Cat stands for Concatenation volume Chang & Chen (2018). G8-C16, G16-C24, and G32-C48 combine Gwc volume and Cat volume Guo et al. (2019). Channel and Dims represent the channel and dimensions of the cost volume, respectively. FLOPs are calculated at a resolution of $544 \times 960$.

| Cost Volume | Dims | Channel | Flops | Params | EPE |
|---|---|---|---|---|---|
| Difference | 3D | - | 38.68G | 2.40M | 1.02 |
| Correlation | 3D | - | 54.99G | 4.01M | 0.81 |
| Interlaced8 | 4D | 8 | 288.52G | 2.83M | 0.70 |
| Gwc8 | 4D | 8 | 70.58G | 2.78M | 0.72 |
| Gwc16 | 4D | 16 | 166.92G | 3.89M | 0.66 |
| Gwc24 | 4D | 24 | 327.13G | 5.75M | 0.63 |
| Gwc32 | 4D | 32 | 551.23G | 8.34M | 0.62 |
| Gwc48 | 4D | 48 | 1191.07G | 15.73M | **0.60** |
| Cat24 | 4D | 24 | 328.97G | 5.78M | 0.65 |
| Cat48 | 4D | 48 | 1192.93G | 15.76M | 0.61 |
| Cat64 | 4D | 64 | 2088.31G | 26.09M | **0.60** |
| G8-C16 | 4D | 24 | 328.96G | 5.78M | 0.62 |
| G16-C24 | 4D | 40 | 841.05G | 11.69M | **0.60** |
| G32-C48 | 4D | 80 | 3239.17G | 39.37M | **0.60** |

for training stereo-matching models. Although five data augmentation techniques—random crop, color augmentation, eraser transform, flip, and spatial transform—are commonly used in stereo matching Lipson et al. (2021); Xu et al. (2023), their empirical efficacy specifically for stereo matching has not been thoroughly explored. This study investigated these data augmentation strategies to address this gap. Most data augmentation strategies, except for the combined use of HSFlip and random crop, lead to a decline in the model's EPE metric of SceneFlow. This is because stereo matching involves pixel-level matching, and these data augmentations (color augmentation and spatial transform) can affect the alignment of pixels. The combination of ColorAug, Erase, and Scale (CES) shows the best generalization performance on KITTI2015, with the lowest EPE of 1.56 and D1_all of 7.64, although it increases the EPE on SceneFlow to 0.7240. These findings underscore the importance of selecting appropriate data augmentation methods to enhance model accuracy and robustness.

### 4.3.2 Feature Extraction

As shown in Table 2, pretraining the backbone is crucial for stereo matching as it enhances the model's ability to extract robust and informative features. Furthermore, the choice of backbone significantly influences the model's performance and computational efficiency. MobilenetV2 Sandler et al. (2018) and EfficientNetV2 Tan & Le (2021) are lightweight CNNs that are particularly efficient in extracting local features, which are crucial for stereo matching. Their designs allow them to perform well with relatively low computational complexity. RepViT Wang et al. (2024a) is a Transformer-based architecture, which excels in capturing long-range dependencies and global context. While RepViT captures global features well, it might struggle with the fine-grained, pixel-level accuracy required for precise stereo matching. MPViT Lee et al. (2022) combines the strengths of both CNNs and Transformers. The CNN components effectively capture local features, while the Transformer components excel in modeling global context. This hybrid approach allows MPViT to leverage the advantages of both architectures, resulting in the lowest EPE. In summary, MobilenetV2 Sandler et al. (2018) offers a good balance for applications with limited computational resources, while more complex architectures like EfficientNetV2 and MPViT provide superior accuracy at the cost of higher computational requirements. To the best of our knowledge, our work is the first to explore the transformer-based feature extraction and the combination of CNN and transformer feature extraction for stereo matching.

Table 4: **Ablation study on SceneFlow Mayer et al. (2016) – Disparity Regression and Refinement.** ArgMin refers to Differentiable ArMin. Context stands for ContextUpasmple Lipson et al. (2021); Xu et al. (2023).

| Regression | Refinement | Flops | Params | EPE |
|---|---|---|---|---|
| ArgMin | None | 58.47G | 2.69M | 0.76 |
| ArgMin | RGBRefine Xu & Zhang (2020) | 117.85G | 2.81M | 0.72 |
| ArgMin | Context | 70.58G | 2.78M | 0.71 |
| ArgMin | Context+RGBRefine Xu & Zhang (2020) | 129.95G | 2.89M | 0.69 |
| ArgMin | Context+DRNetRefine Xu & Zhang (2020) | 129.88G | 2.89M | 0.69 |
| ArgMin | ConvGRU Lipson et al. (2021); Xu et al. (2023) | 3023.88G | 12.51M | **0.46** |

### 4.3.3 COST CONSTRUCTION

In Table 3, an ablation study on various cost volume strategies for stereo matching is presented. For these experiments, one-quarter of stereo image features are used to construct the cost volume. The study begins with simpler 3D cost volume methods: Difference Khamis et al. (2018) and Correlation Wang et al. (2021), yielding higher EPE of 1.02 and 0.81, respectively, at lower computational costs. This suggests that while efficient, these methods may lack the nuanced disparity capture necessary for complex scenes. The Interlaced8 model, introduced by MobileStereoNet Shamsafar et al. (2022), achieves the same EPE comparable to the Gwc8 model. However, its computational expense is substantially higher, with a flop count of 288.52G, significantly larger than that of the Gwc8 model. The group-wise correlation and concatenation models demonstrate a clear trend: as the channel depth increases, the EPE improves, indicating improved disparity estimations through richer feature capture. The combined volume (G8-C16) offers a more optimal balance between computational load and disparity estimation accuracy, which achieves an EPE of 0.62. G16-C24 and G32-C48 do not significantly improve EPE, despite a dramatic increase in computational load, especially for G32-C48, which demands 3239.17Gflops and has 39.37M parameters. These results highlight the delicate balance between accuracy and computational efficiency in designing cost volumes for disparity estimation. While deeper and combined volumes reduce the EPE, the gains might be marginal compared to the significant increase in computational requirements, raising questions about the practicality of these approaches in resource-constrained environments.

### 4.3.4 DISPARITY REGRESSION AND REFINEMENT

The Differentiable ArgMin Kendall et al. (2017); Chang & Chen (2018); Guo et al. (2019); Shen et al. (2021); Xu et al. (2022); Gu et al. (2020); Zhang et al. (2019); Shamsafar et al. (2022); Wang et al. (2021) introduced by GCNet Kendall et al. (2017), calculates initial disparity by converting matching costs into probabilities via softmax and then computing a weighted sum of these probabilities across all disparity levels. As shown in Table 4, various strategies show differing impacts on model performance in this ablation study on disparity refinement for the SceneFlow test datasets. Without refinement, the model has an EPE of 0.76. RGBRefine and Context methods slightly improve EPE to 0.72 and 0.71, respectively, with a modest increase in computational resources. Combining these methods further reduces EPE to 0.69, indicating marginal benefits from their integration. However, ConvGRU refinement substantially improves EPE 0.46, albeit at a significant cost in computational complexity (3023.88 Gflops) and model size (12.51M). This highlights a trade-off between accuracy improvements and increased computational demands.

## 5 A STRONG PIPELINE: STEREOBASE

A strong baseline in deep stereo-matching research is critical for several key reasons. First, it serves as a vital reference point, enabling a clear assessment of new methods against an established standard. Second, a strong baseline allows for precise evaluation of the impact of specific changes, whether they are new data augmentation methods, different network architectures, or innovative disparity estimation techniques. This helps in isolating and understanding the contribution of each component to the overall performance. Additionally, a solid baseline ensures fair and meaningful comparisons across studies, providing a common ground for evaluating different research outcomes. This is crucial for maintaining consistency and validity in comparative analyses. In summary, a strong baseline is

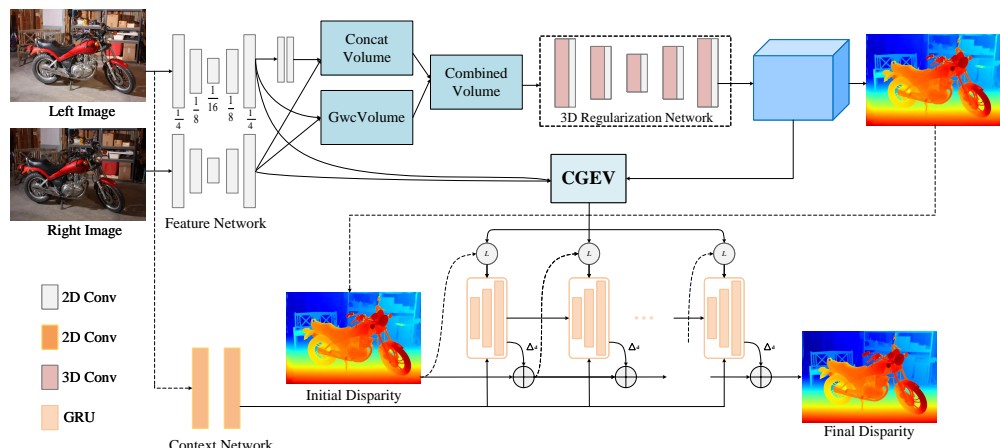

Figure 4: **Overview of our proposed StereoBase.** GwcVolume represents Group-wise correlation volume Guo et al. (2019). CGEV refers to Combined Geometry Encoding Volume Xu et al. (2023).

Table 5: **Results on SceneFlow Mayer et al. (2016), KITTI 2012 Geiger et al. (2012), KITTI 2015 Menze & Geiger (2015) leaderboard, and DrivingStereo Yang et al. (2019b).** All results on DrivingStereo Yang et al. (2019b) are derived using the OpenStereo. Underline refers to evaluation in the non-occluded regions only STTR Li et al. (2021). **Bold**: Best.

| Method | SceneFlow EPE | KITTI 2012 3-noc | KITTI 2012 3-all | KITTI 2015 D1-bg | KITTI 2015 D1-fg | KITTI 2015 D1-all | DrivingStereo EPE | DrivingStereo D1-all |
|---|---|---|---|---|---|---|---|---|
| STTR Li et al. (2021) | 0.43 | - | - | 1.70 | 3.61 | 2.01 | OOM | OOM |
| PSMNet Chang & Chen (2018) | 1.09 | 1.49 | 1.89 | 1.86 | 4.62 | 2.32 | 1.19 | 2.26 |
| GwcNet Guo et al. (2019) | 0.76 | 1.32 | 1.70 | 1.74 | 3.93 | 2.11 | 0.99 | **1.36** |
| CFNet Shen et al. (2021) | 1.04 | 1.23 | 1.58 | 1.54 | 3.56 | 1.88 | **0.98** | 1.46 |
| AANet Xu & Zhang (2020) | 0.87 | 1.91 | 2.42 | 1.65 | 3.96 | 2.03 | 2.91 | 15.16 |
| Mobilestereo-3D Shamsafar et al. (2022) | 0.80 | - | - | 1.75 | 3.87 | 2.10 | 1.06 | 1.61 |
| COEX Bangunharcana et al. (2021) | 0.68 | 1.55 | 1.93 | 1.74 | 3.41 | 2.02 | 1.34 | 2.70 |
| FADNet++ Wang et al. (2021) | 0.76 | - | - | 1.99 | 3.18 | 2.19 | 1.44 | 5.15 |
| CascadeStereo Gu et al. (2020) | 0.72 | - | - | 1.59 | 4.03 | 2.00 | 1.31 | 2.84 |
| IGEV-Stereo Xu et al. (2023) | 0.47 | 1.12 | 1.44 | 1.38 | 2.67 | 1.59 | 1.06 | 1.50 |
| GANet+ADL Xu et al. (2024) | 0.50 | **0.98** | 1.29 | 1.38 | 2.38 | 1.55 | - | - |
| NMRF-Stereo Guan et al. (2024) | 0.45 | 1.01 | 1.35 | **1.28** | 3.13 | 1.59 | - | - |
| Selective-IGEV Wang et al. (2024b) | 0.44 | 1.07 | 1.38 | 1.33 | 2.61 | 1.55 | - | - |
| MoCha-Stereo Chen et al. (2024) | 0.41 | 1.06 | 1.35 | 1.36 | 2.43 | 1.53 | - | - |
| StereoBase(Ours) | **0.34** | 1.00 | **1.26** | **1.28** | **2.26** | **1.44** | 1.15 | 2.19 |

essential for meaningful advancements in deep stereo matching, ensuring that new developments are substantial, accurately assessed, and broadly applicable.

## 5.1 PIPELINE

In light of our comprehensive analysis, the goal of this section is to establish a strong baseline model that surpasses existing standards in performance. StereoBase embodies this objective. As shown in Figure 4, given the left and the right images, the pre-trained MobileNetV2 Wightman (2019) networks are used as our foundational backbone, extracting features at a reduced scale of 1/4th the original size to form the cost volume. The G8-C16 cost volume is utilized to achieve an optimal balance between computational load and disparity estimation accuracy. Hourglass networks Xu et al. (2023) were implemented for cost aggregation, while convGRU Xu et al. (2023) strategies were applied for the final disparity regression.

## 5.2 COMPARISON WITH STATE-OF-THE-ART METHODS

In our comprehensive evaluation, we benchmarked StereoBase against current state-of-the-art methods on SceneFlow Mayer et al. (2016), KITTI2012 Geiger et al. (2012), 2015 Menze & Geiger (2015), and DrivingStereo Yang et al. (2019b) (More implementation details in the Supplementary Material). On the SceneFlow Mayer et al. (2016) test set, we achieve a new SOTA EPE of 0.34.

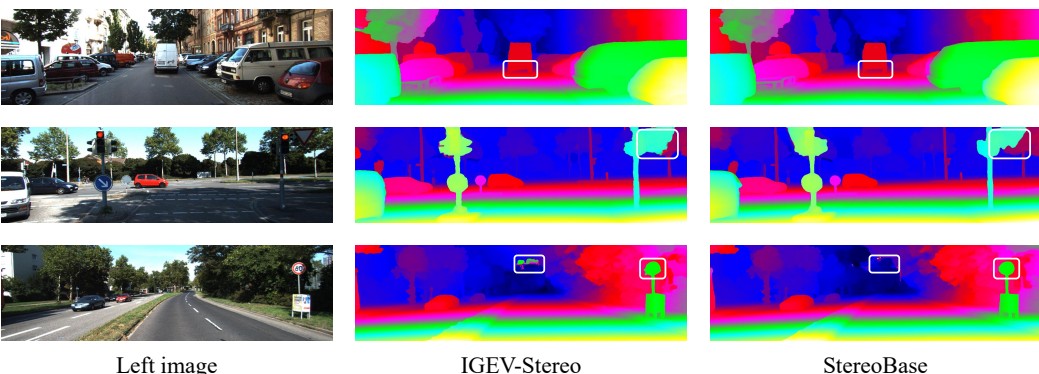

Left image       IGEV-Stereo       StereoBase

Figure 5: **Visualization results on KITTI2015 dataset.**

Table 6: **Cross-domain evaluation** on Middlebury, ETH3D, and KITTI all training sets. All methods are only trained on the Scene Flow dataset. Middlebury is tested on half-resolution. The model with [†] indicates the implementation of OpenStereo. **Bold**: Best.

| Method | KITTI2012 | | KITTI2015 | | Middlebury | ETH3D |
|---|---|---|---|---|---|---|
| | D1-all(%) | EPE | D1-all(%) | EPE | bad 2.0(%) | bad 1.0(%) |
| STTR[†] | 49.72 | 6.80 | 40.26 | 6.16 | OOM | 38.89 |
| PSMNet[†] | 30.51 | 4.68 | 32.15 | 5.99 | 33.53 | 18.02 |
| CFNet[†] | 13.64 | 2.27 | 12.09 | 2.89 | 23.91 | 7.67 |
| AANet[†] | 7.23 | 1.27 | 7.72 | 1.41 | 22.45 | 18.77 |
| Mobilestereo-2D[†] | 18.34 | 2.45 | 21.21 | 2.78 | 34.04 | 13.89 |
| Mobilestereo-3D[†] | 18.96 | 2.79 | 19.69 | 3.40 | 29.32 | 13.71 |
| GwcNet[†] | 23.05 | 2.76 | 25.19 | 3.58 | 29.87 | 14.54 |
| COEX[†] | 12.08 | 1.80 | 11.00 | 2.48 | 25.17 | 11.43 |
| FADNet++[†] | 11.31 | 1.77 | 13.23 | 2.97 | 24.17 | 25.53 |
| CascadeStereo[†] | 11.83 | 1.83 | 12.03 | 2.69 | 27.27 | 11.68 |
| IGEV[†] | 4.88 | **0.98** | **5.16** | **1.18** | **8.47** | 3.53 |
| StereoBase(Ours)[†] | **4.85** | 0.99 | 5.35 | **1.18** | 9.76 | **3.12** |

The quantitative comparisons, as summarized in Table 5, clearly illustrate the edge of StereoBase in handling complex stereo-matching scenarios with greater precision. Further, we submitted our results to the KITTI2012 Geiger et al. (2012) and 2015 Menze & Geiger (2015) leaderboards, where StereoBase outperformed all published methods across all metrics. On KITTI2015 Menze & Geiger (2015), our StereoBase outperforms IGEV Xu et al. (2023) by 9.43% on D1-all metric, respectively. In addition, we evaluate the generalization performance of StereoBase. As shown in Table 6, StereoBase exhibited exceptional performance in a zero-shot setting. This evaluation further validates the adaptability and potential of StereoBase in handling diverse and challenging stereo vision tasks.

# 6 CONCLUSION

This paper introduces OpenStereo, a benchmark designed for deep stereo matching. Our initial endeavor involved re-implementing the most state-of-the-art methods within the OpenStereo framework. This comprehensive tool facilitates the extensive reevaluation of various aspects of stereo-matching methodologies. Drawing on the insights gained from our exhaustive ablation studies, we proposed StereoBase. Our StereoBase ranks 1[st] on SceneFlow, KITTI 2015, 2012 (Reflective) among published methods and performs best across all metrics. In addition, StereoBase has strong cross-dataset generalization. StereoBase not only demonstrates the capabilities of our platform but also sets a new standard in the field for future research and development. Through OpenStereo and StereoBase, we aim to contribute a substantial and versatile resource to the stereo-matching community, fostering innovation and facilitating more effective and efficient research.

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
