# OPENSTEREO: A COMPREHENSIVE BENCHMARK FOR STEREO MATCHING AND STRONG BASELINE

## 1 DATASETS AND EVALUATION METRICS

**SceneFlow** Mayer et al. (2016) is a synthetic stereo collection, counting 35,454/4,370 image pairs for training and testing, respectively. The resolution of SceneFlow is $960 \times 540$ and the ground truth is dense disparity maps. The data is split into two categories: cleanpass and finalpass. Cleanpass refers to the synthetic images that are generated by clean renderings without post-processing, while finalpass images are produced with photorealistic settings such as motion blur, defocus blur, and noise. In evaluations, we utilize the widely-used metrics the end point error (EPE) as the evaluation metrics.

**KITTI.** KITTI 2012 Geiger et al. (2012) and KITTI 2015 Menze & Geiger (2015) are datasets that captured from real-world scenes. KITTI 2012 contains 194 and 195 image pairs for training and testing, respectively. KITTI 2015 provides two sets with 200 image pairs for training and testing. All KITTI datasets provide sparse ground-truth disparities guided by the LiDAR system. For evaluations, we calculate EPE and the percentage of pixels with EPE larger than 3 pixels in all (D1-all) regions. All two KITTI datasets are also used for cross-domain generalization performance evaluation, with EPE and a >3px metric (i.e., the percentage of points with absolute error larger than 3 pixels) reported.

**Middlebury 2014** Scharstein et al. (2014) has two sets of 15 image pairs for both training and testing, respectively. The stereo images are captured from indoor scenes, with 3 resolution options, which are full, half, and quarter. In evaluations, only the training image pairs with half resolution will be used to evaluate the cross-domain generalization performance, and EPE and 3px Error metric are reported.

**ETH3D** Schops et al. (2017) has stereo images collected from both indoor and outdoor environments with grayscale formate. It comes with 27 and 20 image pairs for training and testing, respectively, and a sparsely labeled ground truth for reference. In evaluation, only the training set is used to report the cross-domain generalization performance. And we adopt the EPE and 3px Error as the metric.

**DrivingStereo** Yang et al. (2019) is a large-scale stereo dataset, which contains 180k images covering different weather conditions, such as foggy, cloudy, rainy, and sunny. The DrivingStereo dataset is hundreds of times larger than the KITTI Stereo dataset Geiger et al. (2012); Menze & Geiger (2015).

## 2 MAIN MODULES OF OPENSTEREO

Technically, we follow the design of most PyTorch deep learning projects and divide OpenStereo into three modules, *data*, *modeling*, and *evaluation*.

*Data module* includes three key components: data loader, which is responsible for loading and preprocessing the data; data sampler, which is used to select a subset of data for each iteration; and data transform, which applies various transformations to the data to increase the diversity and complexity of the training set.

*Modeling module* is built upon the `BaseModel` base class, which predefines many behaviors of the stereo matching during both training and testing phases. This module consists of four essential components, namely the *Feature Extraction*, *Cost Construction*, *Cost Aggregation*, and *Disparity Processor*, which are critical to current stereo matching algorithms.

*Evaluation module* is utilized to assess the performance of the obtained model. Recognizing that different datasets require distinct evaluation protocols, we have incorporated these varied protocols directly into OpenStereo, significantly simplifying the evaluation process.

## 3 SUPPORTED METHODS

PSMNet Chang & Chen (2018) is one of the most crucial and frequently cited networks in stereo matching. The official implementation code is available at https://github.com/JiaRenChang/PSMNet.

GwcNet Guo et al. (2019) employs a Group-wise Correlation Stereo Network to exploit both local and global matching cost volumes. https://github.com/xy-guo/GwcNet.

AANet Xu & Zhang (2020) focuses on adaptively fusing multi-scale features through a lightweight and efficient adaptive aggregation module. https://github.com/haofeixu/aanet.

FADNet++ Wang et al. (2021) employs a two-stage architecture, consisting of a feature-metric aggregation stage and a disparity refinement stage. https://github.com/HKBU-HPML/FADNet.

CFNet Shen et al. (2021) utilizes a collaborative feature learning strategy to enhance the representation capacity of features in the matching cost computation. https://github.com/gallenszl/CFNet.

STTR Li et al. (2021) focuses on geometric correspondence problems, including stereo matching, by employing transformers to capture long-range spatial and temporal context information. https://github.com/mli0603/stereo-transformer.

ACVNet Xu et al. (2022) addresses the issue of constructing accurate and efficient cost volumes by adopting an asymmetric strategy. https://github.com/gangweiX/ACVNet.

RAFT-Stereo Lipson et al. (2021) builds upon the RAFTTeed & Deng (2020) architecture, originally designed for optical flow estimation. https://github.com/princeton-vl/RAFT-Stereo.

CoEx Bangunharcana et al. (2021) focuses on context-aware features and cost volume aggregation. https://github.com/antabangun/coex.

CascadeStereo Gu et al. (2020) adopts a cascaded architecture for accurate and efficient disparity estimation. https://github.com/alibaba/cascade-stereo.

MobileStereoNet Shamsafar et al. (2022) builds upon the StereoNet Khamis et al. (2018), which proposes Interlacing Cost Volume Construction. https://github.com/cogsys-tuebingen/mobilestereonet.

IGEV Xu et al. (2023) builds a combined geometry encoding volume that not only encodes both geometric and contextual information but also intricately captures fine-grained details of local matching. https://github.com/gangweiX/IGEV.

## 4 IMPLEMENTATION DETAILS OF DATA AUGMENTATION

For the visualization of the various data augmentation techniques mentioned in Section 4.2.1 of the text, the following examples can be provided.

It is particularly important to note that the disparity map for HFlip appears entirely white because flipping the left and right images in a stereo pair inversely affects the disparity values. When the views are horizontally flipped, the disparity values are multiplied by -1, leading to all disparity values becoming negative. In disparity visualization, negative values typically cannot be represented, resulting in the disparity map appearing as pure white. This phenomenon highlights the importance of considering the geometric relationship between stereo images when applying transformations like horizontal flipping in stereo vision tasks.

The flip data augmentation technique applies HFlips with a probability of 0.5, HSFlips with a probability of 0.5, and VFlips with a probability of 0.1, introducing a variety of orientations in the training data. In parallel, the color data augmentation method randomly applies photometric adjustments either asymmetrically, with a probability of 0.2 to each image in the stereo pair individually, or symmetrically to both images, enhancing the diversity and robustness of the dataset against varying lighting and color conditions. The ScaleTransform data augmentation function scales both images in a stereo pair and their disparity map, applying a base scaling factor with an additional random variation in the x and y dimensions, each with a probability of 0.8, thus introducing variability in size and aspect ratio while adhering to minimum scale requirements. The Shift data augmentation technique randomly shifts the right image horizontally within a range of -10 to +10 pixels, followed by cropping both the left and right images to a predetermined size, thereby introducing spatial variability and

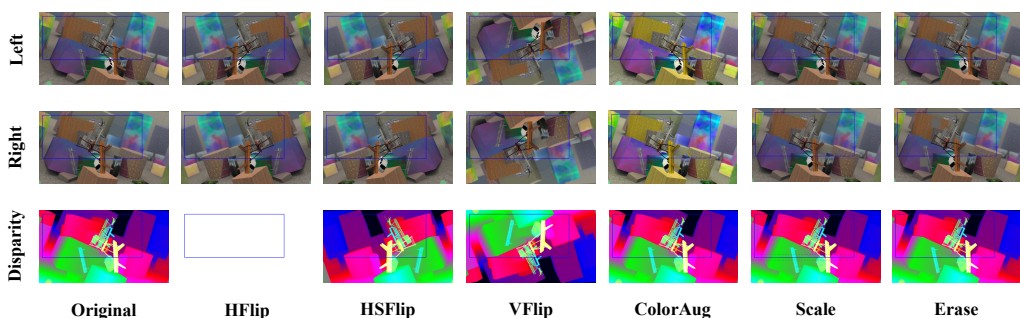

Figure 1: The visualization of stereo images and disparity with different data augmentation. The blue box represents the area where random cropping occurs during training. Notably, when the views are horizontally flipped, the disparity are multiplied by −1, making the disparity map appear pure white.

alignment challenges to the dataset. Lastly, the Random Erase data augmentation, with an eraser probability of 0.5, randomly applies an erasing operation on images, where the dimensions of the erased rectangle vary within a width and height range of 50 to 100 pixels.

Notably, the disparity map for HFlip appears entirely white because flipping the left and right images in a stereo pair inversely affects the disparity values.

## 5 IMPLEMENTATION DETAILS OF STEREOBASE

We set the maximum disparity at 192. Training is driven by the smooth L1 Chang & Chen (2018) loss function. We utilize the AdamW optimizer in conjunction with the OneCycle learning rate scheduler, which adeptly modulates the learning rate to facilitate dynamic adjustments throughout the training phase, enhancing the model's learning efficiency and convergence. For the SceneFlow Mayer et al. (2016) dataset, our training regimen employs a batch size of 16 and a learning rate of 0.0002 over 90 epochs, with the model processing input resolutions set at [320, 736]. On the KITTI dataset, we refine our model by finetuning the pre-trained Scene Flow model using a combined dataset of KITTI2012 Geiger et al. (2012) and 2015 Menze & Geiger (2015) training image pairs. This intensive finetuning process spans 500 epochs, ensuring that the model adapts optimally to the unique characteristics of automotive scenes captured in the KITTI datasets. And we maintain a consistent training setup in terms of batch size, learning rate, optimizer choice, and input resolution as established for the SceneFlow Mayer et al. (2016) dataset. Further adapting our methodology for the DrivingStereo Yang et al. (2019) dataset, we increase the batch size to 48 and adjust the learning rate to 0.0004 over 30 epochs, with the model handling input resolutions of [256, 512].