# OpenReview forum: "OpenStereo: A Comprehensive Benchmark for Stereo Matching and Strong Baseline"
_ICLR.cc/2025/Conference — ICLR 2025 Conference Withdrawn Submission_

### Official Review · Reviewer_bhpT · 2024-10-27

**Soundness:** 2
**Presentation:** 2
**Contribution:** 1
**Rating:** 3
**Confidence:** 5

**Summary:**

The authors propose a comprehensive benchmark focusing on practical applicability for stereo matching. They developed a flexible and efficient stereo-matching codebase called OpenStereo. The codebase includes codes of more than 10 network models. They conducted experiments based on the models that surpassed the results in the original papers. The result of the ablative experiments is StereoBase that ranks first on ScenFlow and KITTI

**Strengths:**

The proposed method OpenStereo ranks first on SceneFlow and KITTI. The proposed framework may be valuable for stereo matching practitioners who want to experiment with different stereo-matching modules.

**Weaknesses:**

1) The paper attempts to provide a benchmark, conduct a survey, and propose a state-of-the-art method for stereo matching, however, it fails to go deep into any of the above-mentioned directions
    - The authors don't provide a benchmark (a dataset with a leaderboard) that other researchers can publish their method and get an evaluation of performance compared to other methods. In practice, the paper relies on existing benchmarks and provides a collection of metrics.
   - The authors attempt to perform a survey and evaluate different components of state-of-the-art stereo-matching methods. The survey includes many important methods but it is not exhaustive and the justification of the choice of the components used to construct StereoBase falls in the category, after an exhaustive search of the 10 implemented models we mixed and matched these components and found that they work best. For example is there a justification why CGEV works better than NMRF?
   - The authors do not go deep into how the proposed framework can be extended and used by researchers to develop novel methods.

The authors need to pick a direction for the paper and go deep in that direction.

2) The paper lacks novelty. The proposed method is a mix and match of prior art.

3) Why the evaluation module only includes submission of the results to KITTI 2012 and 2015?

**Questions:**

N/A

---

### Official Review · Reviewer_MdUT · 2024-11-01

**Soundness:** 3
**Presentation:** 3
**Contribution:** 2
**Rating:** 5
**Confidence:** 4

**Summary:**

This paper introduces "OpenStereo," a tool that aims to streamline the evaluation of stereo matching models by integrating existing datasets and codebases. It also presents "StereoBase," a model designed to perform optimally within this framework. The goal is to provide a standardized comparison of different stereo-matching methods.

**Strengths:**

(1) I appreciate the authors' efforts to integrate existing stereo-matching resources into a cohesive framework, which could facilitate benchmarking and enhance reproducibility within the community.

(2) The detailed experimental setup and comprehensive evaluation are commendable. They provide a clear snapshot of how various established methods perform under a unified protocol. And I believe this paper could serve as a valuable reference for researchers interested in an overview of the stereo-matching landscape.

**Weaknesses:**

(1) The StereoBase, though effective, largely mirrors existing approaches with minimal innovation. Since the paper does not propose new datasets or benchmarks, it mainly refines and combines existing strategies, which might limit its appeal to a readership looking for breakthrough innovations in stereo matching, Therefore, the results are not impressive to me.

(2) The paper presents the StereoBase model as a high-performance solution within the OpenStereo framework, which integrates various existing stereo-matching methods. However, as shown in Table 3 of the manuscript, the computational demands associated with this model are considerable, which could limit its deployment in real-world, resource-constrained environments. To be more specific:
- The group-wise correlation (Gwc) and concatenation volumes used in StereoBase show a drastic increase in computational requirements as the channel depth increases. For instance, the Gwc48 configuration demands 1191.07 Gflops and has 15.73M parameters, which is computationally intensive.
- Similarly, the combined volume configurations like G32-C48 push the computational requirements even further to 3239.17 Gflops with 39.37M parameters. This level of computational complexity is impractical for deployment on embedded systems or mobile devices where power efficiency and processing capabilities are limited.

(3) The paper's comprehensive experiments do not extensively cover the generalization of the proposed model across various real-world scenarios, which might involve dynamic lighting conditions, non-static scenes, or occlusions that are not adequately represented in the benchmark datasets. Since the EPE of the scene flow dataset is already under 1 pixel, improving it from 0.41 to 0.34 may seem like a big improvement since it is too easy to be overfitting for the high overlap of the training and testing scenes. Since the author focuses on `practical` ability, it should stress more on the generation performance in ill-conditioned areas which is more aligned with true practicality. Hence I hope the author shows more specific results in such settings.

There are more datasets with more realistic and higher-quality data such as the booster dataset (https://cvlab-unibo.github.io/booster-web/). FAT dataset (https://research.nvidia.com/publication/2018-06_falling-things-synthetic-dataset-3d-object-detection-and-pose-estimation), distdepth dataset (https://distdepth.github.io/), etc, just to name a few, that is probably more suitable than KITTI and SceneFlow to study stereo (dense disparity maps, more occluded pixels, occlusion masks are given etc). I encourage the authors to not tie your hands tight on these most popular benchmarks. I am not convinced using KITTI and SceneFlow/ ETH3D/Middlebury as the main datasets to study stereo is enough.

**Questions:**

1. The generalization capabilities of StereoBase across different domains and environments are crucial for its application in real-world scenarios. Can the authors extend their experiments to include more diverse datasets that feature challenging conditions such as poor lighting, weather effects, and dynamic obstacles as I believe contained in the drivingStereo dataset?

---

### Official Review · Reviewer_iRgA · 2024-11-03

**Soundness:** 3
**Presentation:** 3
**Contribution:** 3
**Rating:** 5
**Confidence:** 4

**Summary:**

This paper introduces OpenStereo, a benchmark designed for deep stereo matching, sufficient ablation experiments were conducted to verify which configuration was most effective in the stereoscopic matching process. The proposed StereoBase model achieves the SOTA performance.

**Strengths:**

This paper presents a robust baseline that can ensure fair and meaningful comparisons for subsequent improvements.  And through sufficient comparative experiments, tell the community the impact of each part on the final performance.

**Weaknesses:**

1. It is not clear which module is responsible for most of the performance improvement. The result for sceneflow in table 5 is 0.34, but the best result in the previous experiment is 0.46 (table 4). As we all know, in many cases, module A works, module B works, but A+B is not necessarily effective. I think it's hard to tell where the major improvements are coming from just given the results of a final model.
2. The innovation is limited, and I think the framework of StereoBase and IGEV is almost the same, except that the feature extraction part uses MPViT. But that's not an essential contribution either. I think the author should clarify the difference between IGEV and IGEV more clearly.
3. Compared with IGEV, the performance improvement is limited on most real data sets. In table 6, StereoBase and IGEV show similar performance across the most datasets, there is improvement on ETH3D, but there is also negative optimization on middlebury. In the case of vague contribution, I think the improvement is relatively weak.

**Questions:**

1. It is not clear where the major increase came from, and how the final result on sceneflow was increased from 0.46 to 0.34.
2. The innovation is limited, and the performance improvement is limited on real datasets compared with IGEV.

---

### Official Review · Reviewer_DyRW · 2024-11-06

**Soundness:** 2
**Presentation:** 2
**Contribution:** 2
**Rating:** 5
**Confidence:** 4

**Summary:**

The paper proposes a framework for implementing different stereo models and evaluating them in a consistent manner. As part of this effort, the authors implemented a set of stereo models, along with different data augmentation strategies, learning rate strategies and evaluation
THe codebase is based on pytorch and features modular design, includes building blocks for different stereo architectures, supports different evaluation datasets and implementation of some widely used models. The paper provides ablation experiments for data augmentation, learning rate, backbone type, cost volume, disparity regression and refinement. Based on the experiments the paper proposes a stereo model that outperforms other models on some of the datasets.

**Strengths:**

There are many deep learning based stereo models that were proposed over the years and there are several datasets for stereo evaluation. The paper addresses the need for having a unified codebase for training and evaluating different models. This makes the results comparable.

**Weaknesses:**

The paper is mainly proposing a unified codebase to implement different stereo models. While this is an important step to make different stereo works comparable, the paper itself doesn’t provide any new stereo architecture or insight into improving depth accuracy.

The paper doesn’t address the impact of different loss functions and training strategies (other than augmentation and learning rate).

The paper does not provide much detail about the StereoBase model. Architecturally it’s similar to Xu et al’s (2023) IGEV with some modifications based on the observations from the ablation experiments. It’s also unclear how the paper defines a baseline model vs a non-baseline model. Baseline models are typically simple and contain a minimal number of components to perform a task. The proposed StereoBase model is relatively minor modification on a complex stereo model IGEV. Furthermore, the ablation experiment outcomes are limited by the choice of operations, type of dataset, and the metric being optimized. Therefore, it’s unclear how the results from the paper will generalize to new datasets. For example, in Table 5 StereoBase obtains the best performance for SceneFlow but not DrivingStereo.

**Questions:**

What is considered to be a baseline model vs a non baseline model?

Would the ablation experiments using the DrivingStereo dataset change the conclusions?

What type of loss functions were used to train StereoBase? Were all the compared methods trained the same way?

---

### Note · Authors · 2024-11-14

I have read and agree with the venue's withdrawal policy on behalf of myself and my co-authors.